# Improvements in Hydrolytic Stability of Alkali-Activated Mine Tailings via Addition of Sodium Silicate Activator

**DOI:** 10.3390/polym16070957

**Published:** 2024-03-31

**Authors:** Cara Clements, Lori Tunstall, Hector Gelber Bolanos Sosa, Ahmadreza Hedayat

**Affiliations:** 1Department of Civil and Environmental Engineering, Colorado School of Mines, 1500 Illinois Street, Golden, CO 80401, USA; ltunstall@mines.edu (L.T.); hedayat@mines.edu (A.H.); 2Metallurgical Engineering Department, National University of San Agustin de Arequipa, Santa Cataline No. 117, Arequipa 04000, Peru; hbolanos@unsa.edu.pe

**Keywords:** alkali activation, mine tailings, sodium silicate, durability, sustainable construction materials, hydrolytic stability

## Abstract

Over 14 billion tons of mine tailings are produced throughout the world each year, and this type of waste is generally stored onsite indefinitely. Alkali activation is a promising strategy for the reuse of mine tailings to produce construction materials, converting this waste stream into a value-added product. One major problem with alkali-activated mine tailings is their low durability in water (i.e., low hydrolytic stability). In this article, the influence of a mixed sodium hydroxide/sodium silicate alkali activator on the compressive strength, hydrolytic stability, and microstructure of alkali-activated materials (AAMs) were systematically investigated. XRD, FTIR, NMR, and NAD were used to investigate microstructural changes, and a water immersion test was used to show improvements in hydrolytic stability. For gold mine tailings activated with pure sodium hydroxide, the compressive strength was 15 MPa and a seven-day water immersion test caused a strength loss of 70%. With an addition of 1 M sodium silicate in the activator, the AAMs achieved a compressive strength of over 30 MPa and strength loss of only 45%. This paper proposes a mechanism explaining why the strength and hydrolytic stability of AAMs are dependent on the dosage of soluble silicate. A high dosage of sodium silicate inhibits the depolymerization of the source material, which results in a sample with less amorphous aluminosilicate gel and, therefore, lower hydrolytic stability.

## 1. Introduction

Extractive mining produces a large volume of potentially hazardous waste in the form of mine tailings, which are the ground rock and process effluents left over after mineral processing. The mine tailings are usually discharged into ponds contained by earthen dams, where they are stored indefinitely. Although the tailing ponds are intended to safely contain any hazardous materials, the stored mine tailings still can pose an environmental and human health risk as the heavy metals and toxins in the tailings can easily spread to the surrounding soil [1], microbial organisms [2,3] and the environment [4]. In addition, the failure rate of tailing dams is approximately one disaster every eight months [5]. With over 14 billion tons of mine tailings produced globally in 2022 [6], the utilization of this waste stream for the production of value-added products is an exciting opportunity.

Alkali activation is a promising strategy that can utilize the aluminosilicates in the mine tailings to produce a beneficial construction material instead of the tailings just acting as a filler [7,8,9]. The alkali activator is a strongly alkaline solution used to depolymerize the aluminosilicates in the waste material, which produces silicate and aluminate monomers. These building blocks from the source material (plus any extra silica or alumina provided by the alkali activator) then interact and chemically bond to form progressively larger structures through condensation reactions that produce a bridging oxygen molecule. The 3D polymer structure develops through repeated condensation reactions, forming an amorphous aluminosilicate gel. The alkali-activated material (AAM) is then consolidated and cured, often at a slightly elevated temperature (30–80 °C) to accelerate the reaction [10,11]. This dissolution–condensation–curing process is shown in Figure 1.

Mine tailings-based AAMs can exhibit compressive strengths of 5–20 MPa [12,13,14], but the loosely held alkali ions are easily dissolved in water and the aluminosilicate gel can be destroyed by immersion in water. Studies such as [15,16,17,18] have indeed reported poor hydrolytic stability for alkali-activated mine tailings specimens with a strength loss of 70% or more after soaking in water. These studies have cited “incomplete reaction” as the reason for this low integrity in water immersion and suggest adding additional aluminosilicate sources such as fly ash or blast furnace slag to improve the durability properties [17]. These co-binders provide additional silicate and aluminate monomers for reaction, which improves the degree of reaction and, therefore, the compressive strength. According to Wang et.al., increasing the dosage of steel slag from 0–80% increased the compressive strength of geopolymers from 33 to 56 MPa, while the strength retained after 7 days of water immersion was consistently around 50%. While the compressive strength improvements do not directly indicate the degree of reaction, it does act as a proxy measurement because the polymerized aluminosilicate gel is the main strength-contributing phase [19].

Most studies that produce mine tailings-based AAM bricks that meet ASTM standards require a high dosage of co-binders or supplementary cementitious materials [9,13]. While a co-binder can improve the mechanical and durability properties of produced AAMs, adding additional source materials is expensive and runs counter to the goal of immobilizing the highest possible volume of tailings. Instead of using additional aluminosilicate sources, these silicate monomers could be provided via co-activation. Co-activation refers to using a mixed alkali activator composed of an alkali hydroxide and an alkali silicate in solution.

This paper brings together disparate areas of research to explain the co-activation of low-reactivity wastes (such as mine tailings). A previous study found that many different types of mine tailings exhibit less than 5% dissolution of their initial aluminosilicates after 24 h in an alkaline solution [20]. During the same time period, ground granulated blast furnace slag and metakaolin showed about 60 and 80% dissolution of aluminosilicates, respectively. Because mine tailings have a low solubility of aluminosilicates, there is a lack of silicate and aluminate monomers available for reaction. Even if additional monomers are provided via the addition of co-binders or co-activators, the low-reactivity source material can act more as a filler and fails to be well-integrated into the AAM matrix. Therefore, a careful mix design that maximizes the dissolution and polymerization of the source material is critical for the maximum compressive strength and durability of the final AAM.

Previous studies on the use of sodium silicate in the alkali activator have observed that an increase in sodium silicate concentration postpones the strength development of geopolymers, which was attributed to an overly high pH in the system [21]. Another study found that the maximum compressive strength of geopolymers was achieved at a SiO_2_/Na_2_O ratio between 1.0 and 1.26, which is in agreement with other published literature as well [22]. The decline in compressive strength at high dosages of silicate was attributed to hindering the evaporation of water and thus preventing contact between the solids and the activator. However, the specific mechanism of silicate incorporation into the AAM matrix is not well understood. Additionally, the behavior of the mine tailing-based AAM in water is a critical component of performance that has not been adequately addressed in the literature.

This study investigated the activator’s composition, its role in alkali activation, and its influence on the microstructure of alkali-activated materials. The aim of this paper is to understand the mechanism for the incorporation of soluble silicate into the AAM structure and the processes of co-activation that influence the hydrolytic stability of produced AAMs. In this paper, the process–structure–property relationships were explored, and a mechanism is proposed for the incorporation of silicate that accounts for the dosage dependence of the beneficiating effect. Other studies that have prepared AAMs from mine tailings without a high dosage of co-binder (i.e., fly ash, metakaolin, blast furnace slag) have all reported a significant problem with the materials’ durability in water [13,16,17]. This study shows that the addition of sodium silicate to the alkali activator can improve the strength retention after seven days of water immersion, and the subsequent microstructural analyses are intended to explain this improvement.

## 2. Materials and Methods

### 2.1. Mine Tailings

Gold mine tailings were obtained from an artisanal mine near Vitor in the Arequipa region of Peru, which contains about 1.3 million cubic meters of mine tailings deposited in a geomembrane-lined pond. This specific mine is an artisanal site which had no long-term plan for tailings storage or monitoring, so this research was intended to create value-added products using the tailings as the source material. The mine tailings were dried and homogenized prior to their use. Physical characterization tests included: (a) Harvard miniature compaction testing to determine the optimum moisture content and maximum dry density according to US Bureau of Reclamation standard GR-84-14 [23]; (b) sieve analysis and hydrometer testing to determine the particle size distribution according to ASTM C136 and D7928, respectively [24,25]; and (c) Atterberg testing to find the liquid limit, plastic limit, and plasticity index according to ASTM D4318 with hand-rolling and multi-point interpolation [26]. The purpose of plasticity testing was to characterize the clay fraction of the tailings, which affects the swelling behavior of the produced bricks. Based on the results of these tests, the tailings would be classified as AASHTO group A-4 or USCS group SM (silty sand). Figure 2 shows the particle size distribution of the mine tailings along with the results of other physical testing.

Mineralogical composition was determined through X-ray diffraction (XRD). By matching the peak position to the PDF 4+ database from the International Centre for Diffraction Data, the phase composition of the tailings could be determined. The most prevalent mineral in the specimens was quartz, with minor components of albite and muscovite. Diamond powder was used as an internal standard to calibrate the specimen displacement and correct the offset of the peak positions. A more detailed discussion of this treatment and the XRD measurement parameters is presented in Section 2.5, and the XRD pattern is shown in Figure 3.

The chemical composition of the tailings was determined semi-quantitatively using SEM-EDS. The weight percentage of each element was estimated by measuring the quantity of the characteristic X-rays emitted; these values were then averaged over five EDS scans to maximize the accuracy. The oxide composition of the mine tailings is shown in Table 1. X-ray fluorescence was used to confirm the presence of heavy metals such as iron, copper, arsenic, and zinc. However, XRF cannot quantify the lighter elements like Si and Al, so EDS was exclusively used for the quantification. As expected, silicon and aluminum were the most prevalent elements, both of which are crucial for a successful alkali activation reaction [27,28]. The Si/Al ratio of the tailings was 5.6 and the Si/(Al + Fe) ratio was 1.6. The specific contaminants of concern in these gold mine tailings include chromium, arsenic, iron, copper, and zinc.

In addition to the chemical composition of the solids, the soluble fraction of the tailings was characterized using a static leaching test. Ten grams of raw tailings were added to 50 mL of 1 M sodium hydroxide and left to soak at room temperature for 24 h. The liquid was collected and filtered through a 0.45-micron filter then analyzed by ICP-AES. The chemical composition of the soluble fraction in the sodium hydroxide solution (via ICP-AES) is shown in Figure 4. This type of static leaching test has been used to evaluate the amount of reactive silica and alumina in AAM precursors in [29,30].

### 2.2. Alkali Activator

The alkali activator used in this study was composed of sodium hydroxide and sodium silicate mixed in specific molar ratios. A previous study [31] showed that the highest compressive strength for this specific source of mine tailing AAMs was produced with a sodium hydroxide concentration of 10 M. Because the compressive strength of alkali-activated materials generally increases with increasing sodium hydroxide concentrations [32], activator concentrations of 14 M and 18 M were also explored. However, these mix designs yielded samples with large visible cracks after oven curing and excessive efflorescence (i.e., crystal growth on the surface) during water immersion, so they no longer were considered. Therefore, in this study, each activator contained 10 M sodium hydroxide and either 0, 1, or 2 M sodium silicate, termed 10 M, 10 + 1 M, and 10 + 2 M, respectively. The concentration of sodium hydroxide was the same in each activator and the only variable that differed was the concentration of sodium silicate. Not all of the alkali activators react with the mine tailings, as evidenced by the presence of sodium in the leachate after AAMs are immersed in water. However, this concentration of sodium hydroxide has previously been shown to produce the highest compressive strength for mine tailing-based AAMs, so this study continues to use 10 M NaOH as the main alkali activator.

The activators were prepared with ultrapure water (resistance > 18 MΩ), solid sodium hydroxide pellets (NaOH, Fisher Chemical, Hampton, NH, USA, CAS 1310-73-2, anhydrous, reagent grade), and sodium metasilicate powder (Na_2_SiO_3_, Fisher Chemical, CAS 6834-92-0, anhydrous, technical grade). Ultrapure water from a MilliPore MilliQ system was used in place of DI water to improve the purity of the reagents. First, the sodium silicate was mixed with ultrapure water using a magnetic mixer at 200 rpm and 90 °C to aid in dissolution. Once all the sodium silicate was dissolved, the specified quantity of sodium hydroxide was added to the solution. Then, the heating process was stopped, and the solution was mixed for 30 min to ensure complete homogeneity; finally, the activator solution was covered and left at room temperature until it was cool enough to touch. The pH value and ^29^Si NMR spectra of each of the activator solutions are shown in Figure 5.

The peak at −109 ppm is from $Q_{0}^{4}$ four-coordinated silicate groups ($Q_{0}^{4}$), which are present as residual signals from the borosilicate NMR tube [33]. The 10 M activator has the largest peak in this region because there are no silicate groups in the activator. The pure sodium hydroxide does not give off a ^29^Si NMR spectrum, so the background signal is the strongest peak. The 10 + 1 M and 10 + 2 M activator spectra both exhibit two major peaks at −74 ppm and −68 ppm. The −74 ppm peak is from $Q_{0}^{2}$ silicate groups, which are dimers of silica that originated from the sodium metasilicate in the mixture [33]. The peak at −68 ppm signifies $Q^{0}$ silica, monomeric orthosilicate that is another speciation of the sodium metasilicate [33]. The 10 + 2 M activator has much stronger peaks in these two regions because the silicate is more concentrated in this activator. The measured pH is highest for the 10 M activator because it is pure sodium hydroxide with no other ions to interfere. The addition of sodium metasilicate reduces the pH of the activator through weak hydrogen bonding that reduces the activity of the hydroxide groups.

### 2.3. Mine Tailings AAM Preparation

The samples were made using a water-to-tailings ratio of 18%, which is the optimum moisture content for these specific tailing sources. The optimum moisture is defined as the moisture content that allows for most efficient packing of particles and achieves the maximum dry compacted density of a specimen [34]. This study uses the optimum moisture content as the water-to-tailings ratio (w/MT) to decrease microstructural porosity and promote reaction kinetics by increasing the contact between tailings particles and the alkali activator solution. The value of optimum moisture content was determined through Harvard Miniature Compaction testing according to US Bureau of Reclamation standard GR-84-14 [23]. The tailings were compacted at five different moisture contents and a plot was constructed to relate the dry density to the moisture content. The low water content method has been used to produce tailings-based geopolymer bricks in papers such as [13,15,31,35]. By using a low moisture content and pressure compaction to form the bricks, specimens can be rapidly demolded. This method also minimizes the total amount of alkali activator required, providing a more cost effective and environmentally friendly product.

The three different activator compositions/mix designs are shown in Table 2. The mix designs are named according to the alkali activator composition. Activator “10 M” is composed of 10 molar sodium hydroxide. Activators “10 + 1 M” and “10 + 2 M” are composed of the same 10 molar sodium hydroxide plus one or two molar sodium silicate, respectively. The activator and mine tailings were mixed by hand for five minutes until all the activator was absorbed. The mixture was then allowed to stand at room temperature for 15 min to allow all the tailing particles to hydrate and thereby improve homogeneity. Because the application of this research is to produce tailings-based bricks near the mine site in rural Peru, hand tools were used throughout instead of motorized or mechanized components. Stainless steel molds (50 mm × 50 mm × 50 mm cubes) were used for casting. To assist with removing the specimens from the molds, WD-40 was used as a lubricant on the bottom and sides of each mold. Each cube was cast in three layers and a standard Proctor compaction hammer was used for consolidation (as defined in ASTM D698) [34]. The sample was cast using three lifts with 71 tamps each, and a straightedge was used to reach all corners and edges. Additional WD-40 was applied to the corners of the mold in between layers, and the top surface of the cube was struck off with a spatula.

After the compaction, the cubes appeared dense and stiff, and the specimens were able to be immediately demolded and placed on a baking sheet for curing. Previous studies have shown that uncovered curing produces specimens with greater hydrolytic stability than covered curing, despite only slight differences in the overall microstructure; therefore, the specimens were cured uncovered [36]. When 70 °C curing conditions were used for the entire duration, cracks appeared on the surface of the specimens within 12 h due to the rapid evaporation of capillary water. Therefore, the specimens were cured uncovered in an oven at 40 °C for 24 h, followed by six additional days at 70 °C. The lower starting temperature helped to reduce the evaporative cracking caused by the water loss in the beginning stages of curing. After removal from the oven, the specimens were kept in a sealed container to minimize carbonation reactions.

### 2.4. Monolithic Characterization

Compressive strength was measured using an MTS hydraulic load frame in a displacement-controlled loading condition with a rate of 0.21 mm/minute. The maximum load during the test was used to calculate the ultimate stress, and the modulus of elasticity was calculated using the secant method for stresses from 10–60% of the total compressive strength.

The water immersion test measures the hydrolytic stability of the AAMs and can indirectly indicate characteristics of the microstructure, such as permeability. One monolithic cubic sample was placed in 1430 mL of ultrapure water for seven days with no refreshing. Approximately 5 mL of leachate was sampled at 4, 12, 24, 72, and 168 h of exposure for later analysis via ICP-AES.

The degree of reaction test was based on the degree of polymerization test presented by Fernandez-Jiminez et al. in [37] and the improvements suggested by Longhi et al. in [38]. See Appendix A for a discussion of how this test differs from the original and to see the validation/comparison to other methods of quantifying reaction extent. A 2 g specimen of alkali-activated mine tailings was first added to a beaker filled with 250 mL of ultrapure water and magnetically stirred for three hours. The mixture was dried in an oven overnight at 100 °C and then the remaining specimen was added to a beaker with 250 mL of 1:20 dilution of HCl (1.4 M) and magnetically stirred for three hours. Instead of filtering the residue, the solution was progressively diluted with ~2 L of water, and the liquid was decanted. The specimen was dried again in an oven at 100 °C and the mass was recorded. The residue from the HCl step was added to a ceramic crucible and placed in a furnace at 1000 °C for at least 12 h. The mass was recorded again, and the quantity of each phase was determined by finding the percentage of mass loss experienced in each step (water, HCl, furnace).

### 2.5. Microstructural Characterization

The pore structure properties of the geopolymers were analyzed using nitrogen adsorption testing (NAD). A piece of geopolymer brick was ground in a mortar and pestle to a particle size of 0.84–1.2 mm (US sieve #16 to #20). The granulated sample was then dried at 40 °C to remove residual moisture for 24 h, or until the mass taken two hours apart showed no change. The samples were degassed at 70 °C, with unrestricted evacuation (below 10 mmHg) for approximately six to eight hours. The nitrogen adsorption analysis was performed on the degassed sample with an adsorption curve from 0.01 to 0.99 P/P_0_. The Brunauer–Emmett–Teller (BET) method of single-layer adsorption was used to calculate specific surface area, and the pore size distribution was constructed using the Barrett, Joyner, and Halenda (BJH) model on the desorption branch.

Solid-state Fourier transform infrared spectroscopy (FTIR) was used to analyze the chemical bonding structures in the geopolymer samples. The flat detector was used with a diamond top plate, and spectra were collected from 400–4000 cm^−1^ with a spectral resolution of 4 cm^−1^. The data were analyzed using ThermoScientific OMNIC software (version 9.13) and peak positions were compared to reference papers to determine chemical bonding structures within each sample.

XRD diffractograms were collected using a PANalytical PW3040 X-ray Diffractometer with Cu Kα radiation with a voltage of 45 kV and a current of 40 mA at a scan speed of 0.0425°/s. The data were collected from 7° to 90° 2θ at 0.0334° per step. The PDF 4+ database from the International Centre for Diffraction Data was used with PANalytical HighScore software (version 5.1) to determine the constituent phases. The internal standard (synthetic diamond powder) was used to correct for specimen displacement and the zero shift of the peak positions.

### 2.6. Nuclear Magnetic Resonance (NMR)

Although NMR is a very precise and useful technique for assessing the polymerization degree of alkali-activated materials [29,33,39,40,41], mine tailings usually cannot be analyzed by NMR because the magnetic particles in the tailings interfere with the application of the magnetic field. A novel preparation method is used in this paper in order to allow mine tailing-based AAMs to be analyzed via NMR. It can be observed that the entirety of the ferro-magnetic and para-magnetic particles were removed from the raw mine tailings via magnetic separation using a Davis Tube and a wet high-intensity magnetic separator (WHIMS).

^29^Si spectra were obtained at a Larmor frequency of 79.495 using a Bruker Ascend 400WB model NMR spectrometer. The alkali-activated specimens were powdered using a mortar and pestle and loaded into 4 mm zirconium rotors for a triple resonance multi-tuned MAS probe. The peak positions were referenced to pure silica gel as an external standard. Four thousand transients were acquired with a 6 µs pulse delay and a 5 s recycle delay. Five spectra were acquired for each sample and a phase correction was applied to each one, using the background noise as a reference point. The five processed spectra were combined to improve the signal-to-noise ratio using Bruker Top Spin software (version 3.7).

Liquid NMR was used to analyze the alkali activator solutions. ^29^Si spectra were obtained at a Larmor frequency of 59.627 on a Bruker Avance-III 300 NMR Spectrometer. The alkali activator solutions were prepared with deuterated water as the solvent and loaded into a borosilicate NMR tube with a fluoropolymer tube liner to minimize background from the tube. The peak positions were referenced to deuterium in the solvent. Four hundred scans were acquired with a dwell time of 21 µs.

## 3. Results

### 3.1. Monolithic Testing Results

The compressive strength and modulus of elasticity of the alkali-activated mine tailings (50 mm × 50 mm × 50 mm cubes) before and after soaking in water are presented in Figure 6a and Figure 6b, respectively. When bricks were prepared from just tailings and water, the compressive strength was less than 2 MPa (not shown in Figure 6). Therefore, the 10 M NaOH activator caused the alkali activation reaction to occur and produced samples with a strength of 14.6 MPa. Thus, we can conclude that the sodium silicate improved the degree of reaction but is not the only active component of the mixture. Figure 6a shows that the water immersion test caused a strength loss of 70% in the 10 M samples, 53% in the 10 + 1 M samples, and 35% in the 10 + 2 M samples. Generally, samples with a higher compressive strength are expected to perform better during water immersion, as the hydrolytic stability depends heavily on the strength of the matrix and the products of the alkali activation reaction [42,43].

The 10 M and 10 + 2 M specimens had very similar intact compressive strengths (14.6 MPa and 15.2 MPa, respectively) but experienced different levels of disintegration during the water immersion (dissolved strengths of 4.3 MPa and 9.8 MPa, respectively). The AAMs made with the 10 + 1 M activator solution had the greatest compressive strength both before and after the water immersion test and the highest modulus of elasticity before soaking. While all the post-soaking specimens had similar stiffness properties, the 10 + 1 M activator still appeared to produce the most durable specimens after the dissolution test, considering the degree of compressive strength and stiffness retained. The superior performance of the 10 + 1 M activator solution is further discussed in Section 3.3. The 10 + 2 M activator produced the most diverse specimens: for the same preparation process and mix design, four of them had compressive strengths of 9.7, 10.0, 11.7, and 14.3 MPa, and one had an intact strength of 30.3 MPa and a dissolved strength of 15.8 MPa. Since the sample population was small, the high-strength sample could not be discarded as an outlier, thereby contributing to the relatively larger error bars for the 10 + 2 M specimens.

The stiffness (i.e., modulus of elasticity) of the AAM samples followed the same trends as the compressive strength values. The stiffness properties of AAMs depend heavily on the source material used. For mine tailings-based AAMs, [31] reported elastic moduli were 0.4–1.8 GPa for compressive strengths of up to 30 MPa; in metakaolin geopolymers, [44,45] reported elastic moduli were 6–7 GPa and 1.2–6.5 GPa, respectively.

In order to determine the damage mechanism of the water immersion that causes such high strength loss in the AAMs, the leachate from a selected 10 + 1 M cube was sampled. The visual observation of the cubes suggested that the swelling and disintegration of the gel may be a primary damage mechanism, so ICP-AES analysis was used to evaluate whether the aluminosilicate products were solubilized in water.

The results of ICP-AES for the 10 + 1 M specimens during the dissolution test are shown in Figure 7. The data show that the aluminosilicate gel indeed dissolved, and the damage could not be solely attributed to disintegration or swelling of the sample. The concentration of selected elements in the leachate is shown over time in Figure 7, and the general trend is the same for all the elements shown. The deleterious and hazardous elements are shown in the figure by cross markers. These specific elements of concern must be immobilized within the AAM matrix for the safe disposal of the mine tailings. The concentration of each of these elements remained below 0.1 mg/L after seven days of water immersion except for iron and arsenic (8.5 and 0.25 mg/L, respectively). The leachate concentration of chromium was below the detection limit of 0.0045 mg/L.

The silicon, aluminum, and sodium increased in concentration in the leachate very quickly in the beginning of the test, but then plateaued. The sodium was the most soluble in water (likely from excess NaOH dissolving when exposed to water), followed by silicon, sulfur, iron, and aluminum. The presence of aluminum, sodium, and silicon ions in the leachate shows that the gel matrix was dissolved during the water immersion, which was consistent with the observed strength losses (Figure 6).

To determine the reacted proportion of each sample, the degree of reaction test was performed on the raw mine tailings as well as the AAMs prepared with each of the three activator solutions. The mass lost during the water dissolution step was attributed to the less durable products of alkali activation, while the mass lost during the hydrochloric acid dissolution was attributed to the more durable amorphous aluminosilicate gel [37,38]. The mass loss from calcination was also attributed to the aluminosilicate gel and carbonates produced when the material was exposed to the atmosphere. The insoluble residue was primarily composed of crystalline unreacted source material [37].

The results of the degree of reaction testing are shown in Figure 8. The degree of reaction testing on the raw mine tailings (Vitor MTs) demonstrated that 4% of the total mass dissolved in water, 31% dissolved in hydrochloric acid, and 2% of the total mass was lost during calcination in the furnace. These results, therefore, were considered the baseline for the following tests because all the mix designs used the raw tailings as the source material. The remaining sixty three percent of the raw tailings remained as insoluble residue so any improvement (reduction) in this amount was evidence of the reaction and the subsequent transformation of the source material during alkali activation.

In the 10 M sample, the amount of mass dissolved by the water increased to 10% of the total, while the mass lost during the hydrochloric acid step increased to 35%. This result indicates that some of the source material was converted to amorphous aluminosilicate gel (which dissolves in HCl but not water) while some of it was converted to a less durable product that dissolved in water. In the 10 + 1 M sample, there was an 8% mass loss during water immersion. The amount of mass dissolved in the hydrochloric acid increased to 43% for the 10 + 1 M specimens, indicating an increase in the amount of aluminosilicate gel produced as compared to the 10 M specimens and a decrease in the amount of the less durable products. The amount of insoluble residue was the lowest for the 10 + 1 M specimens, which shows that the addition of soluble silicate in the activator caused a greater degree of reaction in the final specimen. The 10 + 2 M specimen had a 12% mass loss during the water dissolution step, which was the highest value for any of the alkali-activated specimens and indicated that this dosage of silicate produced a less stable specimen overall.

The amount of mass lost in the hydrochloric acid dissolution was very similar to the 10 M specimen (34% compared to 35%), which indicated that a similar amount of gel was produced by each of these mix designs. The amount of mass lost in the furnace was also very similar for all three mix designs, which appeared to be an intrinsic property of AAMs that were made from the same source material; for example, mass loss in AAMs exposed to high temperatures has been attributed to the formation of new minerals such as nepheline [46]. Nepheline is an aluminosilicate mineral with the formula Na_3_KAl_4_Si_4_O_16_, which is undersaturated with respect to silica. It forms in alkali aluminosilicates exposed to temperatures of 1000 °C, and the amount of nepheline formed depends on the silicon content of the source material [47]. Despite similarities in their phase composition and the degree of reaction, the microstructure of the 10 M and 10 + 2 M samples in this study were distinct, which explains the difference in performance observed during and after the dissolution test.

### 3.2. Microstructural Testing Results

The pore size distribution constructed from nitrogen adsorption analysis is shown in Figure 9. The summary statistics from the NAD analysis are shown in Figure 10, including the specific surface area and total pore volume. AAMs are considered mesoporous materials, which contain pores primarily between 20 and 500 Å [48]. The “hump” observed in the pore size distribution for the 10 M and 10 + 2 M dissolved specimens illustrates this phenomenon well. The voids within the matrix of the AAMs indicate that there were gaps between the nanoprecipitates that make up the aluminosilicate gel [49]. Therefore, an increase in the total pore volume indicated that the aluminosilicate gel in this study had been dissolved or damaged, increasing the average distance between each nanoprecipitate component. Similarly, an increase in the specific surface area indicated that the solid fraction of the matrix had become more porous, and the extent of the voids had increased correspondingly.

The change in the pore characteristics of the specimens during water immersion depends on the concentration of sodium silicate in the alkali activator. The 10 + 1 M specimen showed a dense microstructure and low porosity both before and after the water immersion, but the 10 + 2 M specimen had a significant increase in total pore volume after the dissolution testing. Both activator formulations had the same effect on the initial pore structure of the AAM because there was a specific maximum amount of silicate that could be incorporated into the matrix. For the 10 + 1 M activator, the source material plus the activator provided this amount of silicate, and, thus, the degree of reaction increased (see Figure 8). For the 10 + 2 M specimens, the excess silicate from the activator was not incorporated into the matrix because the source material did not fully depolymerize. Adding increasingly more soluble silicate became less effective because the high concentration in the activator prohibited the full participation of the source material in the alkali activation reaction.

In the intact (pre-dissolution) state, the 10 M specimen had the largest total pore volume, followed by 10 + 2 M, and the 10 + 1 M specimen had the smallest total pore volume. The BET surface area of the intact specimens followed the same trend: 10 M > 10 + 2 M > 10 + 1 M. Because voids can be interpreted as gaps between structural precipitates, this result indicated that the 10 + 1 M specimen had the densest microstructure and the greatest quantity of aluminosilicate gel and precipitates before dissolution. This result suggests that the addition of soluble silicate in the activator solution produced a greater volume of solid aluminosilicate gel (also seen in Figure 8), and that changing the dosage of silicate produced different microstructures within the specimens. This result was consistent with the observed increase in compressive strength for the 10 + 1 M and 10 + 2 M intact specimens. The mechanism for this strength improvement is discussed in detail in Section 3.3.

In all three mix designs, an increase in the void space was observed after the dissolution test. In the 10 M and 10 + 2 M specimens, in particular, the dissolution of the aluminosilicate gel produced a large number of new mesopores, which can be seen in the “hump” shape of the graph (dissolved specimen—blue line). This result indicates that the spaces between the aluminosilicate nanoprecipitates increased. Because the volume of the specimens did not change appreciably, it can be inferred that this increase was due to the dissolution process. While each of the specimens showed an increase in void space (and, thus, a decrease in aluminosilicate gel precipitates), the effect was most pronounced for the 10 + 2 M specimens. The total pore volume increased by more than four times after the dissolution test, ultimately surpassing the total pore volume of the dissolved 10 M samples. This result indicates that the initial pore characteristics of the materials were not sufficient indicators of performance during the dissolution test. The effect of the sodium silicate concentration on the micro- and macro-scale observations is discussed in more detail in Section 3.3.

The FTIR spectra of intact and dissolved AAMs is shown in Figure 11, with the peak locations and associated bonding structures presented in the inset table. The sharp band at 1426 cm^−1^ indicated the stretching of the –CO_3_ and Ca–O bonds [50,51,52]. The presence of CO_3_ can be attributed to the formation of calcite, which is mainly composed of calcium carbonate. This reaction can happen when an AAM specimen is exposed to carbon dioxide in the atmosphere. The fact that calcite is easily dissolved upon immersion in water explains the absence of this peak in each of the dissolved spectra. The band at 880 cm^−1^ can be attributed to C–O stretching in sodium carbonate, which forms in the same way as calcite [53]. This compound also dissolves easily in water, which explains the absence of the 880 cm^−1^ peak in the dissolved specimens’ spectra. Because all the specimens were exposed to the same environmental conditions, atmospheric carbonation was not expected to contribute to the observed differences among the specimens.

Various FTIR peaks can be attributed to the bonding structures of aluminosilicate gel, and there are certain signatures that suggest that a material has undergone alkali activation. The formation of alkali–activated products can also be observed in the SEM micrographs presented in Figure 12. The left image shows the raw mine tailings particles, which appear as discrete angular particles on a black background (carbon tape). The right image shows the hardened geopolymer product. The amorphous structure of the aluminosilicate gel appears as a glassy phase with no defined edges, which can be seen surrounding the unreacted tailings particles. This indicates that the source material dissolved to provide monomers for the polymerization reaction. Even the unreacted tailings particles appear to be well-integrated into the AAM matrix, which is ideal for long-term durability and strength.

The main signature of successful alkali activation in the FTIR spectrum is the nonappearance of a band in the region of 920 cm^−1^ and the occurrence of another one in the range of 690–696 cm^−1^ [54]. These bands indicate that the Si–O–Si and Si–O–Al bonds are present and that all the octahedral aluminates have been consumed, showing that the source material has been transformed into new aluminosilicate structures through the incorporation of bridging oxygens [54]. Peaks in the region of 975–990 cm^−1^ (observed at 982 in this material) are indicative of the asymmetric stretching of the Si–O–Si and Al–O–Si bonds in the gel matrix [30,55,56]. The peaks between 775 and 798 cm^−1^ indicate the symmetric stretching of the Si–O–Si bonds in aluminosilicates, which have been observed at 795 and 775 cm^−1^ in these materials [57]. Additionally, the peak at 694 cm^−1^ can be attributed to Al–O–Si and Si–O–Si bending [30,53,55], and the peak at 1162 cm^−1^ can be attributed to partial substitution of Al in the aluminosilicate gel structure [58]. Researchers have reported this substitution to be up to 15% of the total bonding in the gel [58]. Because FTIR only shows the chemical bonds present in a material (not the coordination states or short-range ordering), these results confirmed that the mine tailings underwent alkali activation but do not confirm the extent of polymerization reaction in the AAM.

Nuclear magnetic resonance (NMR) was used to show the coordination states and bonding structure of the silicon atoms in the AAM matrix. The spectra for the raw tailings and all three AAMs are shown in Figure 13. The peak frequencies and widths for the various coordination states of silica are well documented in the literature [33]; the Q^0^–Q^3^ peaks are located at −72, −81, −89, and −97 cm^−1^, respectively. The NMR results show that the spectra were similar for the raw tailings and each alkali-activated specimen, with slight differences due to the increased incorporation of Q^2^ and Q^3^ coordinated silica in the 10 + 1 M and 10 + 2M samples. Each spectrum exhibited a sharp peak between −80 and −81 cm^−1^, which corresponded to the Q^1^ coordination state of the silica, meaning that each silicon atom was incorporated into the matrix by only one bridging oxygen, and the other three bonding sites were occupied by an –OH group. This result indicates that the polymerization process was incomplete since the characteristic cross-linkages require higher coordination states of Q^3^ and Q^4^ [59,60]. Since the peak is very sharp, it does not appear to be composed of multiple overlapping signals, and, therefore, deconvolution was unnecessary.

Even though the peak frequencies were similar between the raw material and the AAMs, the full width at half the maximum intensity (FWHM) was greater for the raw mine tailings. This result indicates that the silicate structures in the tailings were bulkier than in the AAMs, which caused a longer relaxation time and, thus, a broader peak [39]. The sharpness of the AAM peaks indicated that the aluminosilicate source material was dissolved into smaller oligomers with shorter relaxation times during alkali activation [61]. The change in peak shape shows that the alkalination and depolymerization steps were completed (the necessary precursors for alkali activation), but the remaining four steps for a full polymerization reaction (gel formation, polycondensation, networking, and solidification) did not occur [39]. Instead, the strengths of these AAMs developed through the nanoprecipitation of the aluminosilicate oligomers and the solidification of the N-A-S-H gel [62]. While the aluminosilicate species did not create a polymer network (as in geopolymerization), the AAMs with good compressive strength and hydrolytic stability properties were created through alkali activation with 10 M sodium hydroxide and 1 M soluble sodium silicate.

While NMR analysis is critical for directly quantifying the degree of reaction in a potential geopolymer specimen, it is often impossible to use this technique to evaluate the extent of the alkali activation of mine tailings since this source material usually has magnetic particles. This study demonstrated that it is possible to evaluate magnetic precursor materials with NMR by following the demagnetization pretreatment discussed in the methodology section. It also highlights why it is imperative to employ NMR analysis for assessing polymerization. While the structural differences between AAMs and geopolymers cause them to have different monolithic behavior, one cannot accurately determine whether a material is a true geopolymer from monolithic testing methods alone; in this study, this may have led to the erroneous conclusion that the superior performance of the 10 + 1 M specimen was due to its higher degree of polymerization. Ultimately, atomic-scale analysis is necessary to determine the connectivity of the aluminosilicate gel network and to evaluate the total extent of reaction.

### 3.3. Effect of Sodium Silicate Concentration on Microstructure

When the source material (raw mine tailings) was dissolved by the alkali activator, the chemical bonds were hydrolyzed (attacked by water) and the alumina was released in the first step. The release of the alumina produces a silica-rich surface precursor, which is then further dissolved to release the silicates [63]. In the sodium hydroxide-only mix design (i.e., 10 M), the hydrolysis of the chemical bonds in the source material continued and the silicates and aluminates that were released reacted to form longer aluminosilicate chains. This was a condensation reaction, where a new bridging oxygen was formed from two hydroxide (–OH) groups and a water molecule was released. This process is shown in the top third of Figure 14 (10 M activator).

When soluble silicate is present in the activator, there is already monomeric silica available for reaction before the source material begins to dissolve. As soon as the alumina from the source material was released, the soluble silica reacted to form small aluminosilicate chains. The silicate from the source material was slower to dissolve, but as hydrolysis continued, both the soluble silicate and the source material silicate were incorporated into the gel structure through condensation reactions. The final AAMs contained soluble silica in the matrix, as well as silicates and aluminates from the source material. The 10 + 1 M specimens were more durable because there was sufficient monomeric silica available for reaction, but not enough silica to retard the dissolution of the source material. This process is shown in the middle third of Figure 14 (10 + 1 M activator).

In the 10 + 2 M specimens, the monomeric silica reacted with the source material aluminates in the same way. However, because the concentration of soluble silica was higher, it also interfered with the dissolution of the source material. Once the alumina was released, the aqueous silica could form new Si–O–Si bonds with the silicates in the source material, which slowed the hydrolysis of the chemical bonds and the dissolution of the aluminosilicates [63]. This did not allow all the source material to be broken down into silicate and aluminate monomers, which prohibited a full reaction and slowed the formation of the alkali-activated material. This result is shown in the bottom third of Figure 14 (10 + 2 M activator).

The hydrolytic stability of the 10 + 2 M specimens was much poorer than that of the 10 + 1 M specimens, which shows that soluble silica can improve the performance of AAMs only when it is applied in the correct dosage. Too much soluble silica in the activator can be a detriment to the structure of the aluminosilicate gel and the overall performance of these materials. While additional soluble silica can improve the compressive strength and hydrolytic stability of the material, excess silica retards the dissolution of the source material, decreasing the final material’s performance. Despite the compressive strength improvements for intact 10 + 2 M specimens, their water immersion performance was poor. Therefore, the dosage of silica must be carefully controlled to allow the source material to fully dissolve and participate in the condensation reactions in order to produce a material that has sufficient quantities of aluminosilicate gel (and, thus, high compressive strength) while maintaining a high degree of reaction of the source material (and, thus, good stability during water exposure).

## 4. Conclusions

This paper studied the alkali activation of gold mine tailings without an added aluminosilicate binder. The influence of a mixed sodium hydroxide/sodium silicate alkali activator solution on the compressive strength and hydrolytic stability of mine tailings AAMs was investigated. The strength retention after water immersion was optimized by manipulating the sodium silicate dosage to maximize the participation of the source material in the alkali activation reaction. Based on these results, a mechanism for the incorporation of soluble silicate into the microstructure of AAMs was proposed.

AAMs created with sodium hydroxide activator showed poor durability in water. The microstructure of the AAMs became more porous during water immersion, the compressive strength greatly decreased, and there was a visible deterioration and swelling of the specimens. The ICP-AES results indicate that these effects were attributable to the dissolution of aluminosilicate gel.The hydrolytic stability improved with the addition of sodium silicate to the activator. AAMs made with the combination sodium hydroxide/sodium silicate activator (10 + 1 M activator) exhibited a denser microstructure (lower porosity both before and after water immersion). The stiffness of the intact specimens increased as well as the compressive strength.At higher dosages of silicate (10 + 2 M activator), the monomeric silicate reacted with the source material to retard dissolution, which caused the specimen to have a lower degree of reaction than for the optimal activator concentration (10 + 1 M activator) and, therefore, poorer durability during water immersion.

When designing AAMs, especially for source materials with low reactivity (e.g., mine tailings) it is important to utilize soluble silica in the activator to improve the pore structure of the final product. However, the dosage of silica must be carefully optimized to ensure the complete dissolution of the source material so that the degree of reaction is maximized and thereby produces the strongest and most durable specimens with the greatest amount of aluminosilicate gel.

## Figures and Tables

**Figure 1 polymers-16-00957-f001:**
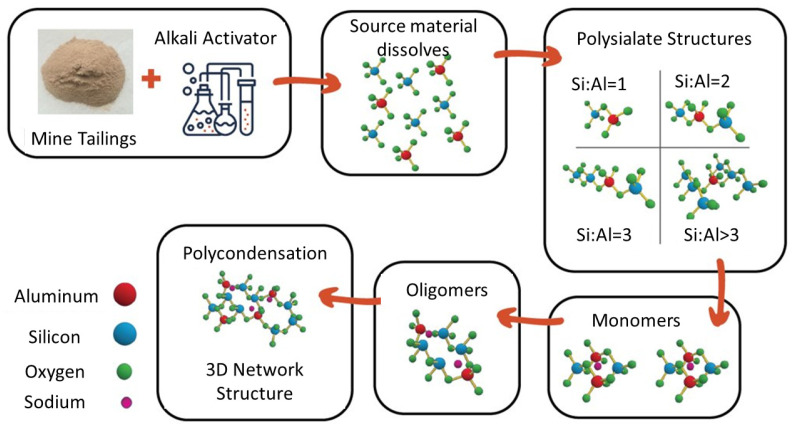
Molecular model of source material dissolution, condensation, and hardening of final AAM structure.

**Figure 2 polymers-16-00957-f002:**
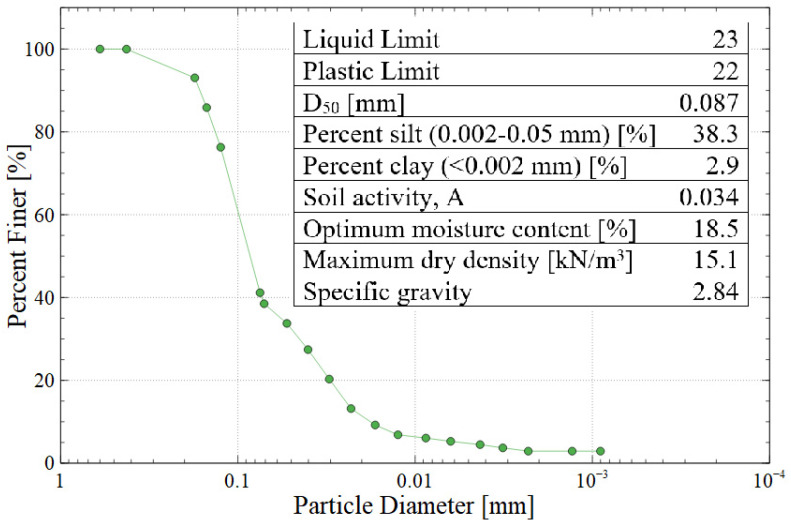
Particle size distribution and summary of the physical properties of Vitor mine tailings (AASHTO A-4 or USCS SM).

**Figure 3 polymers-16-00957-f003:**
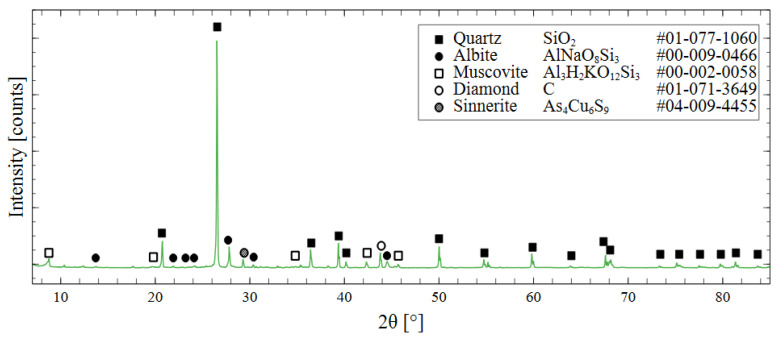
XRD spectrum of raw mine tailings with phase identification using ICDD PDF 4+ database.

**Figure 4 polymers-16-00957-f004:**
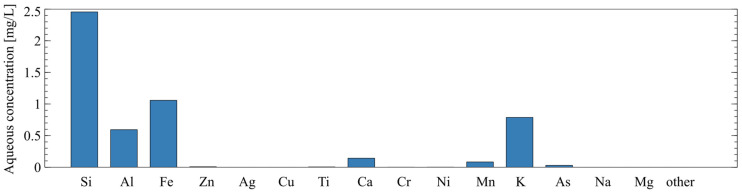
Chemical composition of soluble fraction of mine tailings as determined by static leaching and ICP-AES.

**Figure 5 polymers-16-00957-f005:**
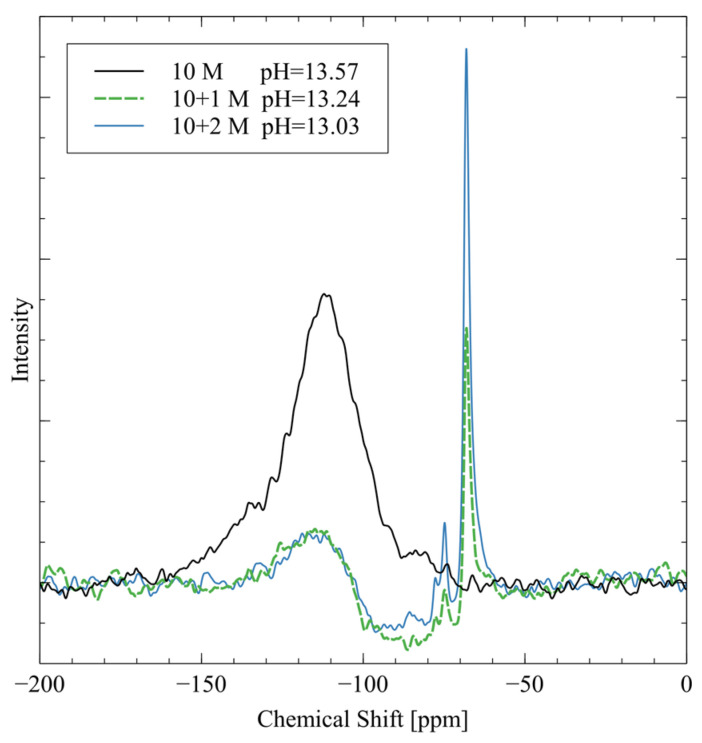
^29^SI NMR spectra and pH values of each alkali activator solution.

**Figure 6 polymers-16-00957-f006:**
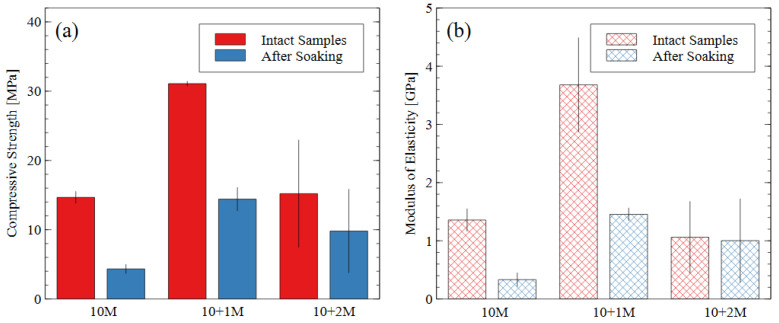
(**a**) Compressive strength and (**b**) modulus of elasticity data for each mix design, before and after water immersion. Error bars represent one standard deviation for tests carried out on three samples (after soaking) or five samples (intact).

**Figure 7 polymers-16-00957-f007:**
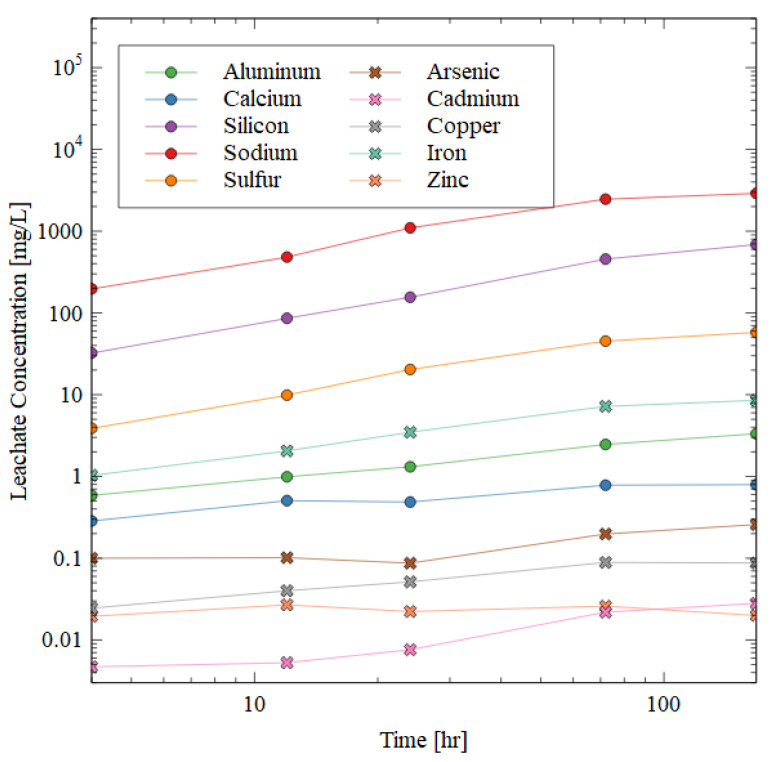
ICP-AES results of the leachate after water immersion of 10 + 1 M specimen. Circular markers indicate elements highly prevalent in the AAM matrix, while crosses indicate deleterious elements that must be immobilized for the successful disposal of mine tailings.

**Figure 8 polymers-16-00957-f008:**
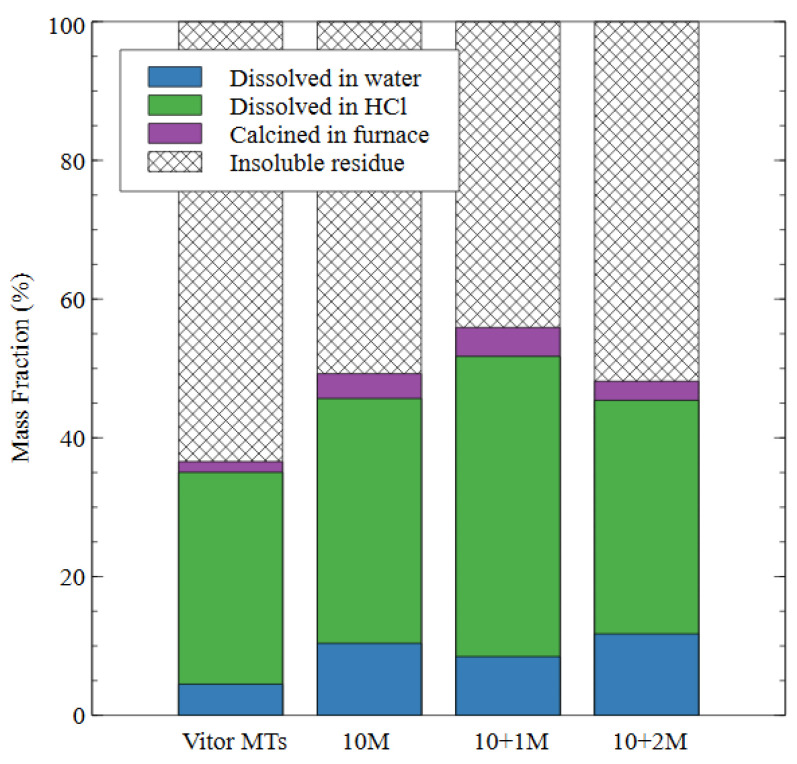
The effect of different alkali-activated mix designs on the degree of reaction of the alkali-activated mine tailings samples.

**Figure 9 polymers-16-00957-f009:**
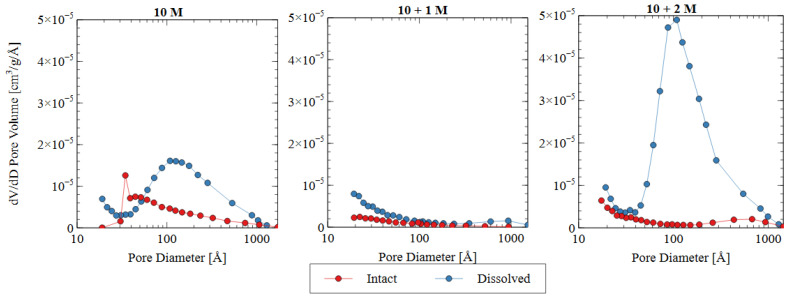
Pore size distributions based on nitrogen adsorption data for three different activator solutions.

**Figure 10 polymers-16-00957-f010:**
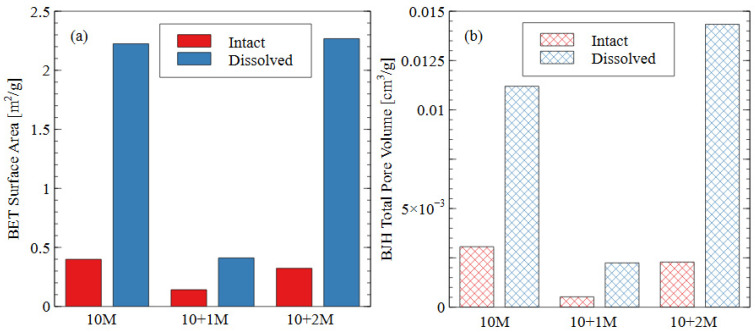
Summary statistics from NAD analysis: (**a**) specific surface area as calculated using BET adsorption theory; (**b**) total pore volume calculated using BJH desorption curve.

**Figure 11 polymers-16-00957-f011:**
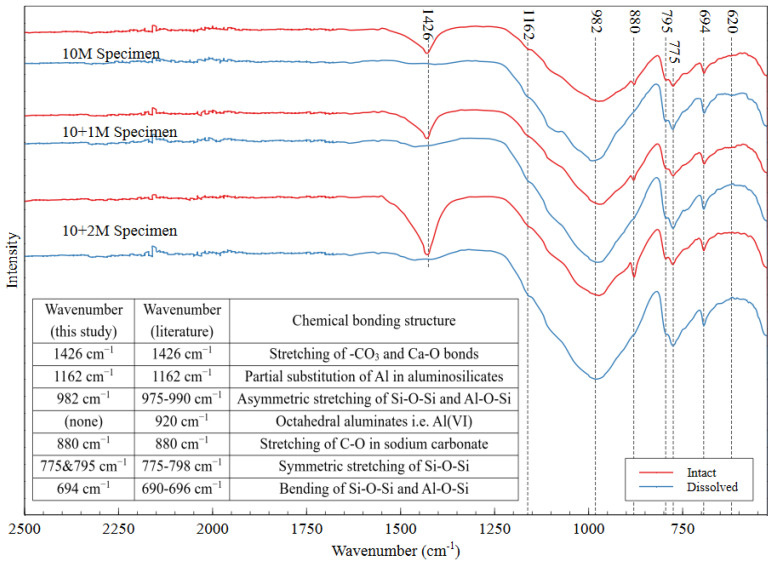
FTIR spectra of intact and dissolved AAMs for three different mix designs, which confirm that the mine tailings underwent alkali activation.

**Figure 12 polymers-16-00957-f012:**
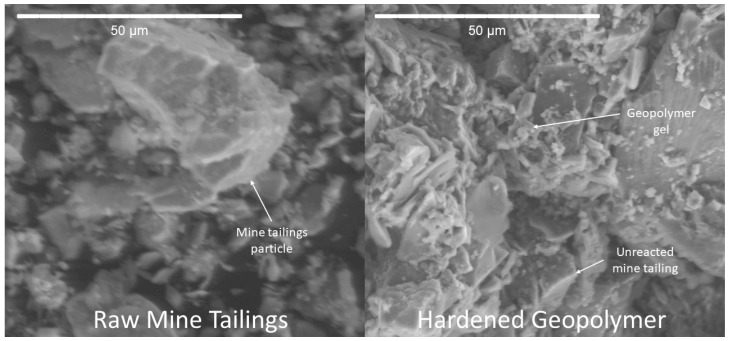
SEM micrographs of raw mine tailings and hardened geopolymer sample.

**Figure 13 polymers-16-00957-f013:**
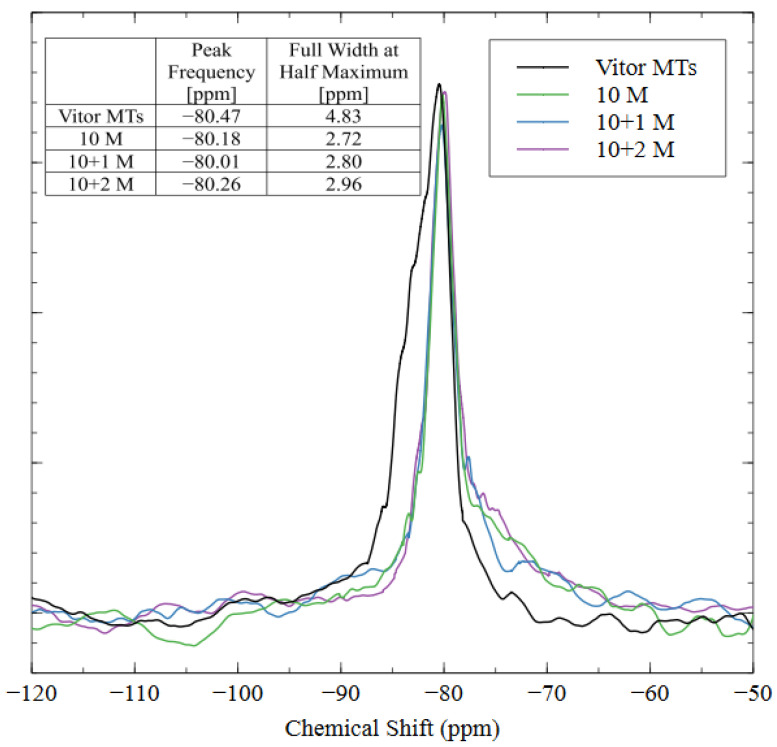
NMR spectra of mine tailings source material and alkali-activated material with three different mix designs.

**Figure 14 polymers-16-00957-f014:**
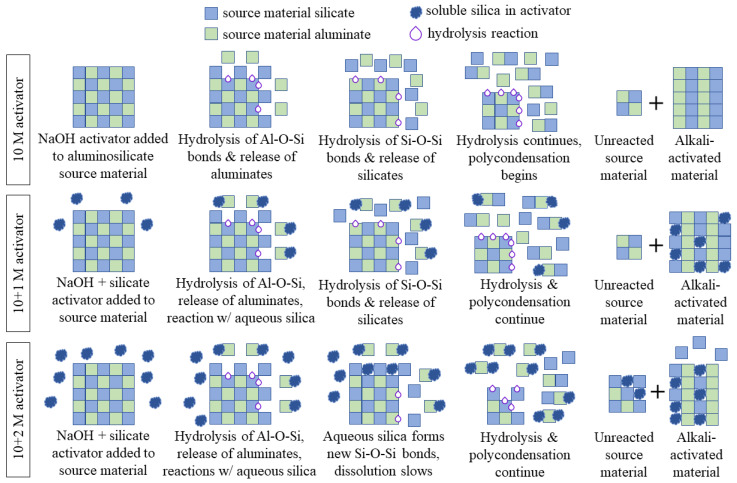
Visual representation of the proposed mechanism of sodium silicate interaction for each of the alkali activator solutions. The 10 + 1 M activator improves the durability of AAMs because it provides monomeric silica for the condensation reactions, but the 10 + 2 M activator has so much excess silica that it reacts with the source material to retard its dissolution. A plus sign in the figure indicates that the system is a mixture of two components.

**Table 1 polymers-16-00957-t001:** Oxides composition of Vitor gold mine tailings.

SiO_2_	Fe_2_O_3_	Al_2_O_3_	K_2_O	MgO	As_2_O_3_	Na_2_O	ZnO	CuO	TiO_2_	NiO	CoO	Other
55.1%	24.6%	9.8%	2.8%	1.1%	1.0%	0.9%	0.8%	0.7%	0.6%	0.5%	0.5%	1.6%

**Table 2 polymers-16-00957-t002:** Mix designs for each of the three activator solutions.

Mix Design Name	Mine Tailings [g]	Water [mL]	Sodium Hydroxide [g]	Sodium Metasilicate [g]
10 M	750	135	54	0
10 + 1 M	750	135	54	16.47
10 + 2 M	750	135	54	32.94

## Data Availability

All data, models, and code generated or used during the study appear in the submitted article.

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
