# Peer review of "Improvements in Hydrolytic Stability of Alkali-Activated Mine Tailings via Addition of Sodium Silicate Activator"

_polymers, 2024, doi:10.3390/polym16070957_

Round 1

Reviewer 1 Report

Comments and Suggestions for Authors

Introduction

Well written introduction

Materials

Figure 2;

Atterberg limits; the liquid limit is within the range of the plastic limit. This material has no plasticity as is evident by the plasticity index of 1.  Very low clay content (as your tailings sample) always means no plasticity.

Readers will be interested to know the reason why a tailings such as your sample which contains lots of coarse solids and is very low in clay content has to be contained in tailings dams as it should dewater and allow dryland creating readily. Please include the reason for such a “tailings issue” with this mine when the solids appear to pose no such problem. I see no reason why your tailing sample to have sufficient moisture content to remain as “tailings fluid”.

For tailings disposal alkali concentrations of 10M and higher are too concentrated to be acceptable. The scheme could work on the bench but it may be inapplicable for real world tailings disposal.

It is not clear how concentration is defined in the paper. Are these concentrations calculated based on the water mass of the tailings or combined (solids +liquid) of the tailings or only the solids content of the tailings? The concentrations the authors are referring to appears to refer “stock solution preparation” of the NaOH and sodium silicate. Please be clearer on the tailings treatment concentrations.

How does the word “efflorescence” in line 169 mean anything?

Is it possible that all the action is due to the sodium silicate?

Compressive strength measurement sample size:  50x50x50 mm cubes, right? May help to remind readers.

Figure 6; Has the number of duplicate measurements been stated to gauge the “importance of the error bars”?

Is it assumed that all the NaOH and silicate react with the solids? This cannot be true and has this been covered in the paper?

Figure 11 is a clear way of displaying the spectra.

Discussion section

Some of the statements in this section were already given in the results section. Same points are repeated in the discussion section as in the results section. You can shorten the paper by having a results and discussion section and avoid repeating the same points.

Conclusion section (the most disappointing section of the paper)

It is way overboard. Everything is not important to get included in the conclusion section. What are the important points of the work? All of it! This section rehashes all the points in the discussion and the results section even experimental procedures such as the “ferromagnetic removal procedure”.  It is so long it makes it meaningless. Just include the CONCLUSIONS.

To the Editor: The paper is well written. I presume the objective of the work is relevant although my reading does not sway me in it. The length of the conclusion section is "unlike any I often come across". 

Author Response

Thank you for your review. Please see attachment.

Reviewer 2 Report

Comments and Suggestions for Authors It is recommended to add a table with the oxide composition of the tailings used to the section “2.1 Mine Tailings”;   Line 128. Please, explain, what does “q” mean in abbreviation “qXRD”?   Line 175. Please explain why the activators were prepared with ultrapure water? How did you get ultrapure water? Lines 228–231. Please indicate what size samples were prepared for this study?   The authors note that one of the objectives of this study was the immobilization of specific contaminants such as chromium, arsenic, iron, copper, and zinc. Therefore, to confirm/refute the fact of increasing the efficiency of “binding” pollutants in geopolymers, it would be great to add data on the chemical/elemental composition of water extracts from the optimal composition of geopolymer composites based on aluminosilicate enrichment tailings as for the initial components (similar to Figure 4). This comment is a recommendation only.

Author Response

(The authors gave the same response as above.)

Reviewer 3 Report

Comments and Suggestions for Authors

This paper discussed the improvements in hydrolytic stability of alkali activated mine tailings via a mixture addition of sodium silicate and sodium hydroxide activators. This study is good at the utilization of second resources like mine tailings. But the content of this paper is not new and creative enough, and there are many obvious errors in the whole paper. So I suggest it is rejected. The detailed comments are as the following:

1. AAM, that is, geopolymer, has been studied several dozens of years. Many solid waste, especially those whose main contents are aluminosilicate minerals, are the focused research objectives. The synthesis mechanism of AAM are explored deeply, including various methods of NMR, FTIR, et al which are also used by this paper. The key point of this paper is NMR results, according to the authors repeated expression. But the authors did not discuss the creativeness. In fact, although from Fig.13, it is difficult to identify Q2 and Q3 clearly, there is difference between these three curves if carefully observed. The compressive strength increases mostly because of the existence of Q2 and Q3.

2. Abstract part, the mine tailings are stored as hazardous material, this is not accurate since most taillings are general waste.

3. Introduction part should introduce the research progress on the AMM produced by activated gold mine tailings since many research have been conducted to explore the behaviour of gold mine tailings and performance of the product.  

4. The pH of 10M sodium hydroxide is 10.57, which is not aligned with the common sense.

5. The chemical compositions of mine tailings are not right. In general, Al, Si, Fe, et al are all in the forms of oxides or complexes. Thus, the total mass percentage of these elements will much more than 100% according to the data of Figure. 4. In addition, ~5% Ag and ~5% Cu in the tailings are too high.

6. Figure 7 showed that the sodium is the most soluble, not silicon.

7. How can the geopolymer gel be identified in Figure 12? 

Author Response

(The authors gave the same response as above.)
